# Design and fault diagnosis of DCS sintering furnace's temperature control system for edge computing

Na Qu[1,2], Wen You[1] *

**1** Department of Mechanical and Electrical Engineering, Changchun University of Technology, Changchun City, China, **2** Department of Electrical Information Engineering, Jilin University of Architecture and Technology, Changchun City, China

* youwen@ccut.edu.cn

**Data Availability Statement:** All relevant data are within the paper and its Supporting Information files.

## Abstract

Under the background of modern industrial processing and production, the sintering furnace's temperature control system is researched to achieve intelligent smelting and reduce energy consumption. First, the specific application and implementation of edge computing in industrial processing and production are analyzed. The industrial processing and production intelligent equipment based on edge computing includes the equipment layer, the edge layer, and the cloud platform layer. This architecture improves the operating efficiency of the intelligent control system. Then, the sintering furnace in the metallurgical industry is taken as an example. The sintering furnace connects powder material particles at high temperatures; thus, the core temperature control system is investigated. Under the actual sintering furnace engineering design, the Distributed Control System (DCS) is used as the basis of sintering furnace temperature control, and the Programmable Logic Controller (PLC) is adopted to reduce the electrical wiring and switch contacts. The hardware circuit of DCS is designed; on this basis, an embedded operating system with excellent performance is transplanted according to functional requirements. The final DCS-based temperature control system is applied to actual monitoring. The real-time temperature of the upper, middle, and lower currents of 1# sintering furnace at a particular point is measured to be 56.95°C, 56.58°C, and 57.2°C, respectively. The real-time temperature of the upper, middle, and lower currents of 2# sintering furnaces at a particular point is measured to be 144.7°C, 143.8°C, and 144.0°C, respectively. Overall, the temperature control deviation of the three currents of the two sintering furnaces stays in the controllable range. An expert system based on fuzzy logic in the fault diagnosis system can comprehensively predict the situation of the sintering furnaces. The prediction results of the sintering furnace's faults are closer to the actual situation compared with the fault diagnosis method based on the Backpropagation (BP) neural network. The designed system makes up for the shortcomings of the sintering furnace's traditional temperature control systems and can control the temperature of the sintering furnace intelligently and scientifically. Besides, it can diagnose equipment faults timely and efficiently, thereby improving the sintering efficiency.

**Funding:** This work was supported by Jilin Province Science and technology development plan project in 2020 (No.20200403131SF).

**Competing interests:** The authors have declared that no competing interests exist.

## Introduction

Industrial processing and production involve various equipment and thousands of key components. The failure of any component will severely impact the entire industrial production and cause huge production losses [1]. The normal operation of processing and production equipment and the interaction between components require real-time status detection of the equipment. Because traditional manual detection methods have low efficiency and large errors, sensors are added to detect key components' status and provide real-time feedback for intelligent automated control of industrial production [2–4]. Automated control is based on data calculation. In the industrial Internet of Things (IoT), cloud computing technology based on big data is widely applied. At the current stage, smart industrial production plants depend on the data collected by sensors on industrial equipment; these data are reported to the cloud for calculation and processing by the processor, and solutions are obtained and downloaded in real-time [5]. However, computer technology has been developing rapidly; the Fifth-Generation (5G) and even the future Sixth-Generation (6G) bring much more equipment on the network edge; thus, the exponentially increased network data require powerful calculation methods. Edge computing is a computing center between cloud servers and edge equipment, responding quickly to requests from edge equipment [6, 7]. Therefore, the equipment can be controlled via edge computing during industrial processing and production, thereby promoting the efficient progress of industrial production.

In the metallurgical industry, powder metallurgy materials have always been indispensable industrial materials in medicine, railways, and military fields [8]. The sintering furnace is the central equipment in the powder metallurgy industry, a piece of high-energy-consuming equipment that makes the powder compact to obtain the required mechanical properties and microstructure via sintering [9]. Due to the high requirements for intelligent equipment control based on edge computing, the temperature control of sintering furnaces is also a vital link worthy of in-depth study. A sintering furnace is a furnace for bonding powder material particles at high temperatures to forge denser composite materials. The powder metallurgy production process includes material mixing, forming, sintering, and reshaping [10]. Sintering is the most crucial link to overall energy consumption.

Meanwhile, the temperature distribution during sintering will directly affect the final sintering quality. Therefore, precise control of the sintering furnace temperature is the basis for completing the powder metallurgy sintering task. The Distributed Control System (DCS) adopts distributed control and centralized operation and management. It combines computer technology, system control technology, communication technology, and multimedia technology to control the advanced and complicated laws.

To improve the production efficiency of industrial products, the industrial field has gradually begun to attach importance to edge computing and put it into use in actual production. The idea of the edge computing intelligent body is introduced based on DCS. The Programmable Logic Controller (PLC) is adopted for the automatic control of the sintering furnace temperature. A fault prediction model based on fuzzy logic is proposed to accurately diagnose the faults of the sintering furnaces, and timely adjust the temperature of the combustion chambers, ensuring that the temperature of the sintering furnace is controlled within the target value range. Furthermore, the sintering furnace's temperature control system can maintain excellent and stable performance. Innovatively, advantages of edge computing, namely the capability of real-time processing and latency reduction, are organically combined with industrial intelligent equipment for the intelligent control of production. According to the process and characteristics of the sintering furnace, the PLC technology and configuration software are

organically combined to develop a set of an automatic temperature control system for the sintering furnace suitable for modern industrial production.

## Related work

Sintering ignition is a vital step in the sintering process and the starting point of the entire combustion process. In the case of unstable air pressure, stabilizing the ignition temperature of sintering has significant economic and scientific value. Du et al. [11] proposed an intelligent control strategy based on the ignition temperature prediction for the ignition process of iron ore sintering. Combining the mechanism analysis method and the data-driven method, they established an ignition temperature prediction model. Then, the control mode was determined according to the actual operating experience. The intelligent controller for the ignition temperature was designed to obtain the required gas flow, and the gas flow controller was used for stabilization. To improve the quality of Selective Laser Sintering (SLS) components and simplify their application process, temperature fluctuations must be resolved. Phillips et al. [12] introduced a method to actively control the laser flux on the powder surface based on infrared temperature measurements. Controlling the energy input by the laser could achieve a high degree of control over the temperature of the final component, whose uniformity could be improved as well. For specimens of constant cross-section, the standard deviation of ultimate flexural strength increased by 45%.

Temperature maintenance is a crucial problem in current automatic control systems. KIlyushin and Golovina [13] developed a comprehensive technique for nonlinear controllers to stabilize the temperature field of the controlled object. The final controller allowed the creation of an adaptive control system to maintain the temperature field. Temperature process control was an indispensable component in the industrial engineering field. Generally, industrial processes can be simulated through Proportional-Integral-Derivative (PID) controllers and various adjustment methods. Nazarudin et al. [14] designed a process control experiment and a primary temperature control process experiment. The Ziegler-Nichols process response curve was applied to adjust the controller, and the final gain and response of the temperature control were analyzed by analyzing the proportional band, integral band, and derivative band values to make a decision on PID adjustment.

The above analysis reveals that to improve the performance of the sintering furnace in the metallurgical industry, the temperature should be controlled through the predictive model to ensure the regular operation of the sintering furnace. However, the process is often complicated. Therefore, in the present work, edge computing, an emerging technology to intelligent industrial equipment, is explored to improve the system's data analysis and processing capabilities, thereby providing accurate prediction and control of the temperature of the sintering furnace.

## Materials and methods

### Industrial processing and production intelligent equipment for edge computing

Edge computing technology is on the rise; its significant developments will significantly impact the infrastructure, networks, and analytics. Edge computing brings processing to the equipment or gateways on the network. The basic concept is driven by the following ideas: (1) particular types of processing must be performed with minimal latency to feedback the processes, such as local analysis, robot functions, and sensor operations. (2) Powerful edge equipment and gateways can compress data for transmission to the cloud, perform preprocessing or

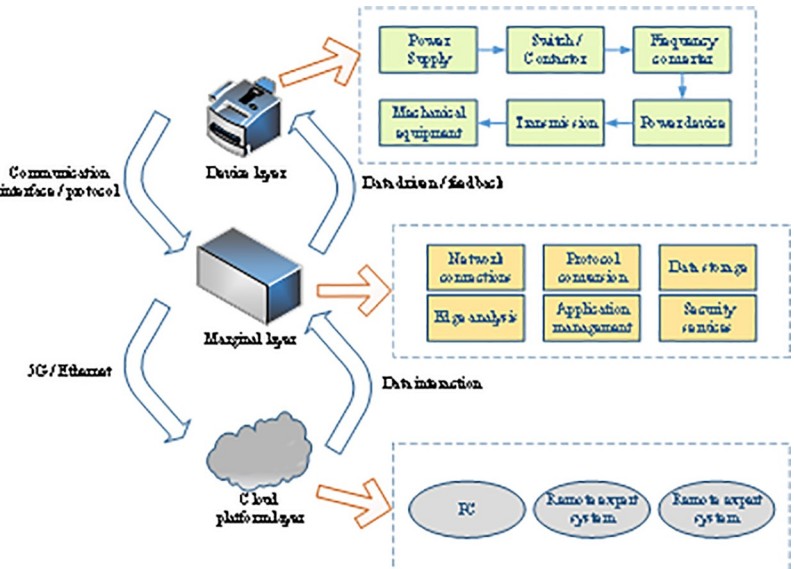

**Fig 1. The overall architecture of industrial processing and production intelligent equipment based on edge computing.**

processing, and coordinate autonomous tasks without having to access a central computer. To better formulate the standards for edge computing, the European Telecommunications Standards Institute revised the name of "mobile edge computing" workgroups with industry specifications into "multi-access edge computing" [15–17].

The interaction between cloud computing and factory equipment has slow information transmission and high bandwidth. Introducing edge computing into the cloud computing-based industrial intelligent equipment architecture can deploy the cloud's core capabilities to the terminal, allowing industrial equipment to respond quickly and efficiently. Edge computing is closely connected to the continuous development of IoT and the launch of 5G mobile networks [18]. Analysis and data may encounter major new opportunities and challenges. Hence, supporting infrastructure must be established, new security requirements must be put forward, and new models are needed to process IoT data. The constructed industrial processing and production intelligent equipment architecture is divided into three layers: the equipment layer, the edge layer, and the cloud platform layer, as shown in Fig 1.

The equipment layer is responsible for the perception, acquisition, and monitoring of primary data, including the status of industrial processing and production equipment and the operating environment. The equipment layer is the key to status perception and control implementation in the intelligent equipment architecture. Different processing equipment and components include power supplies, controllers, inverters, power and transmission devices, and other necessary devices [19]. The equipment layer may not have computing capabilities; nevertheless, it must accurately perceive and collect data via the equipment layer to ensure the operation of the control system. The edge layer includes network interconnection, protocol conversion, data storage, edge computing, and data analysis [20]. In an edge computing network, each layer can receive the intelligent equipment data collected by sensors in its upper layer, process heterogeneous data upwards, schedule different computing tasks, and deploy the algorithm library at the cloud platform layer. If an abnormal status of the equipment or component is detected at the edge layer, the edge layer will issue an adjustment command to the component after the current status is calculated. A remote expert system is deployed at the

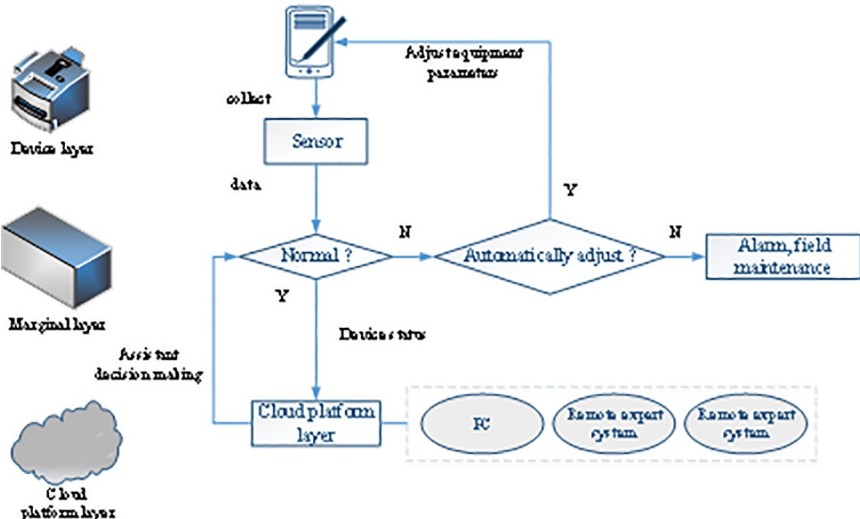

**Fig 2. Intelligent equipment workflow based on edge computing.**

cloud platform layer to provide a rich knowledge system and experiences, make in-depth reasoning and decision-making on the analysis results, and process various issues effectively. Moreover, the cloud platform layer can display the status data of intelligent equipment and components in real-time, and users can openly and transparently understand the overall status of industrial processing and production [21]. The cloud platform can also share real-time data to the mobile PC via the Web terminal, ensuring that the industrial processing and production information can be viewed at any time.

While processing initial redundant data, the cloud platform based on edge computing shortens the distance between the equipment layer and the computing center, saves bandwidth resources, and improves the operating efficiency of the intelligent control system. The edge computing technology is applied to the intelligent equipment architecture, and the isolated initially equipment information is integrated. The specific workflow is presented in Fig 2.

## The overall architecture of the sintering furnace's temperature control system

A sintering furnace is a heating tool composed of a furnace body, a furnace bottom, a vacuum system, a heating mantle, and a temperature control system. The sintering furnace control system regulates temperature, pressure, and vacuum. The overall structure of the control system is displayed in Fig 3, including control cabinets, heating cabinets, and transformer cabinets. In

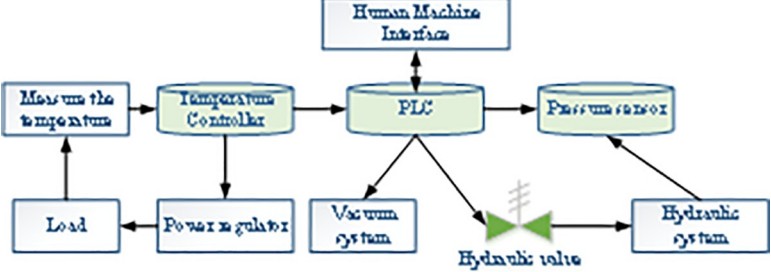

**Fig 3. Logical relationship of control system.**

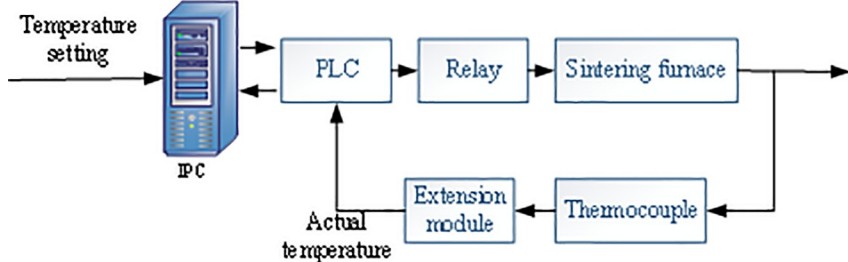

**Fig 4. Three control temperature characteristic curves.**

terms of temperature control, four-way separated control is adopted to avoid the uneven temperature field of the sintering furnace as much as possible due to the relatively large heat dissipation of the furnace roof, furnace door, and furnace body. The current output from connecting the high-precision digital meter FP93 and the thermocouple is employed as the control signal. The heater is controlled by regulating the output power of the power regulator. In terms of pressure control, the pressure signal collected by the pressure sensor is transmitted to the PLC controller through the transmitter for closed-loop Proportion Integration Differentiation (PID) control [22, 23]. In terms of vacuum degree control, the vacuum system and PLC controller are used simultaneously to roughly extract the sintering furnace first and then perform refined extraction after heating and pressuring specific requirements.

The sintering furnace's temperature control system adopts closed-loop control, takes the expected sintering furnace temperature as the set value, converts the feedback information measured by the temperature sensor into a digital quantity, compares the difference between the quantity of preset temperature and feedback temperature, and sends the result into the PID controller. Afterward, a voltage signal is output to control the relay after arithmetic processing, thereby controlling the furnace temperature [24, 25]. The sintering furnace's temperature control system is illustrated in Fig 4. It uses a thermocouple extension module that integrates temperature acquisition and data processing, with automated linearization processing and cold junction compensation functions.

The workpiece undergoes three processes in the sintering furnace. The system controls the three-stage temperature inside the sintering furnace: heating and temperature rising—heat preservation—heating termination. Each step should meet the technical requirements of process production control [26]. First, the sintering furnace must allow three control temperature characteristic curves for processing and production, as presented in Fig 5. The sintering furnace adopts the electric heating sintering method.

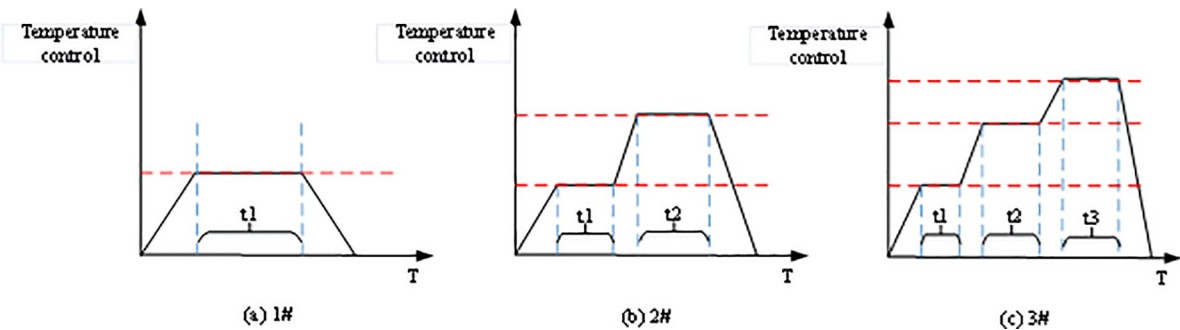

**Fig 5. Three control temperature characteristic curves.**

Moreover, the temperature is monitored at the three positions (upper, middle, and lower) of the furnace to achieve the equilibrium of furnace temperature. However, due to the electric heating source's different heat transfer characteristics at the three points in the furnace, control measures must be taken to avoid significant differences in the three points' rising temperature values. During the temperature rising stage, an isothermal point can be set. Thus, the three points' rising temperature can reach an equilibrium between the temperature differences at the isothermal point. The compulsory adjustment method that reduces the overshoot temperature and the incremental PID control method can be employed to meet automatic temperature control requirements during the heat preservation stage.

## The DSC-based temperature control system

DCS is a comprehensive control system that conforms to the requirements of modern large-scale industrial production automation. It takes the microprocessor as the core and combines multi-disciplinary expertise, such as computer technology, system control technology, and multimedia technology. The information of each scattered point is collected and concentrated via the data channel, and the temperature of the sintering furnace is controlled through monitoring and operation [27, 28]. DCS is presented in Fig 6, including an industrial computer and PLC controller in the control room, temperature controller, communication between industrial computer and PLC (using PrefilBus communication bus), and communication between PLC and temperature controller (using 485 communication bus). The PLC controller's internal circuit in DSC adopts advanced anti-interference technology, with extremely high reliability and scalability [29]. In the system using PLC as the controller, the electrical wiring and switch contacts are significantly reduced compared with the relay system of the same scale. Hence, the fault rate of the system will also be reduced.

The open-loop prediction model of the management layer is integrated into the industrial computer. The open-loop prediction model of the management layer is expressed as follows:

$$u(k) = \frac{1}{\hat{h}_1} \left\{ y_r(k+1) - \sum_{j=2}^{N} \hat{h}_j \cdot u(k+1-j) \right\} \qquad (1)$$

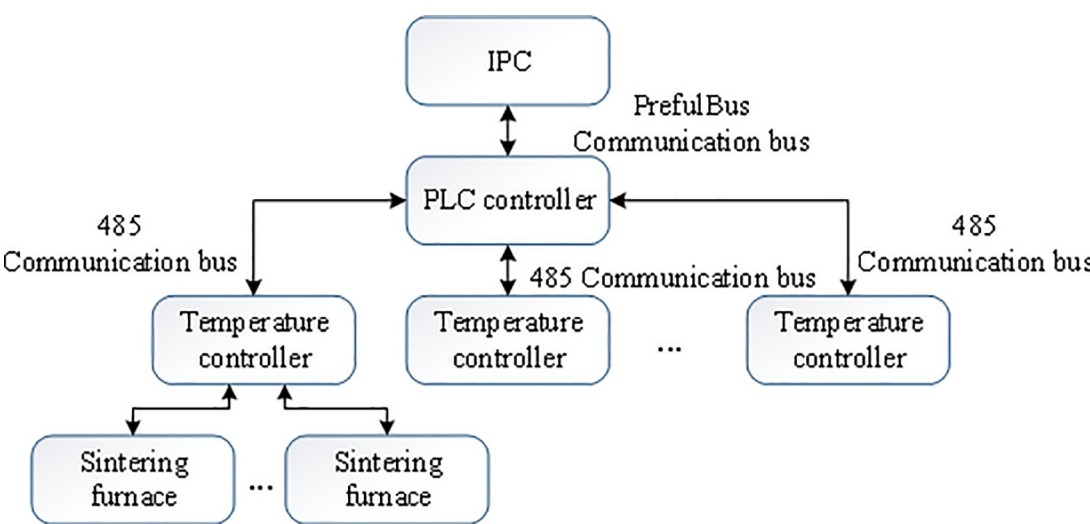

**Fig 6. Three control temperature characteristic curves.**

In (1), $u(k)$ refers to the control increment at time $k$, $y_r(k+1)$ represents the reference trajectory at time $k+1$, $h_j$ describes the temperature value collected at time $j$ in the internal model, and $u(k+1-j)$ signifies the control sequence applied before time $k$, including the control quantity applied before time $k$, and the control quantity to be obtained at time $k$ and thereafter.

On each temperature controller, a closed-loop prediction model and a fault prediction model of the field control layer are integrated; the expression of the closed-loop prediction model of the field control layer is described in Eq (2):

$$u(k) = \frac{1}{\hat{h}_1}\left\{ y_r(k+1) - [y(k) - y_m(k)] - \sum_{j=2}^{N} \hat{h}_j \cdot u(k+1-j) \right\} \tag{2}$$

In (2), $y(k)$ represents the output measurement of the actual process at time $k$, and $y_m(k)$ refers to the output value of the prediction model at time $k$.

## Fault diagnosis and fault-tolerance control of sintering furnace

Components of the sintering furnace to be diagnosed include the detection channel, the sensors, the communication channel, the implementation channel, and the sintering furnace. The fault detection procedure is as follows: (1) determining whether the detection channel and sensor are faulty as per the temperature value output by the temperature controller; (2) detecting whether the communication channel is faulty using the delay query method; (3) performing unit step perturbation on the fault prediction model, and detecting whether the execution channel and the sintering furnace are faulty as per the difference between the output change of the fault prediction model and the actual change [30]. This system introduces fuzzy logic into the parameter-changing process of the sintering furnace. It also combines expert domain knowledge rules to predict the status of the sintering furnace comprehensively. The diagnostic structure of the system is displayed in Fig 7.

(1) Determining whether the detection channel and sensor are faulty: the detection channel is the data transmission path between the sensor and the temperature control device. Whether the temperature value output by the temperature control device is the value under the open-circuit status and the short-circuit status is determined. If the judgment result is "Yes," the detection channel and the temperature sensor are faulty.

(2) Detecting whether the communication channel is faulty: "communication channel" refers to the communication path from each temperature controller to the industrial

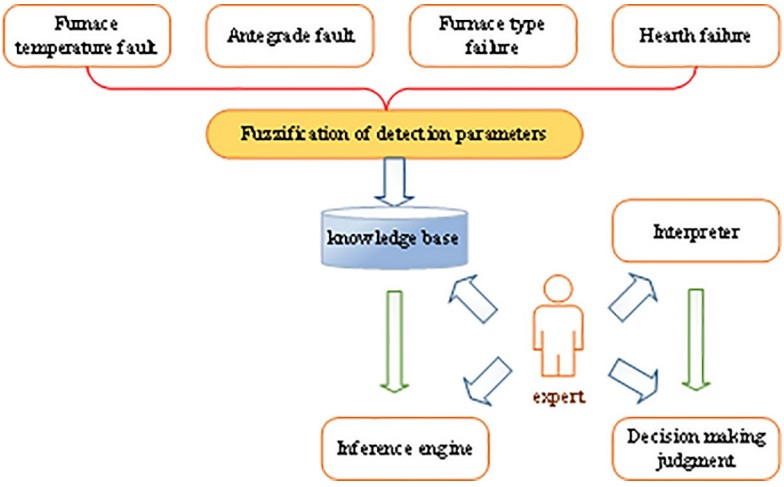

**Fig 7. Diagnostic structure of sintering furnace's fuzzy expert system.**

computer. It detects the time to transmit the test signal from each temperature controller to the industrial computer. If the transmission time is greater than the set time threshold, the communication channel will be judged to be faulty. Otherwise, the communication channel is normal.

(3) Detecting whether the execution channel and the sintering furnace are faulty: if the amount of change output by the fault prediction model and the actual amount of change are greater than the set threshold, the implementation channel and the sintering furnace are faulty.

Assuming that a temperature controller in the sintering furnace DCS controls four sintering furnaces at most. An internal model for the temperature controller to control four sintering furnaces, an internal model for the temperature controller to control three sintering furnaces, an internal model for the temperature controller to control two sintering furnaces, and an internal model for the temperature controller to control one sintering furnace are established, respectively. The temperature controller that controls four sintering furnaces is taken as an example for the detailed description. Suppose the Analog Current (AC) voltage is applied to four sintering furnaces in the offline status. In that case, the temperature values are collected sequentially when these sintering furnaces are online according to the set sampling cycle until the temperature of the collection point reaches the steady-state position. The steady-state position indicates that the temperature of the collection point has reached the preset temperature value. The N temperature values collected at time N constitute the internal model of the sintering furnace. The internal model is stored in the database of the industrial computer as a table. Data collection in the online state is that the industrial computer automatically completes the above tasks through the sampling program. The internal model of each sintering furnace is acquired under the condition that the test conditions are almost the same, thereby reducing the error of the fault prediction model.

A DCS for sintering furnace temperature is constructed, and system fault diagnosis and fault-tolerance control are achieved based on this system. Under the condition of not increasing any hardware cost, the sintering furnace system's fault self-diagnosis and fault-tolerance control functions are achieved through the integrated software algorithm in the system, thereby reducing the system maintenance personnel requirements and maintenance costs and improving the reliability of the sintering furnace system simultaneously [31, 32]. The expression of the fault prediction model is shown in Eq (3):

$$y_m(k + i) = \sum_{j=1}^{N} \hat{h}_j \cdot u(k + i - j) \tag{3}$$

In (3), $N$ represents the cut-off step length, $y_m(k+i)$ refers to the output value of the prediction model at the time $(k+i)$, $h_j$ describes the temperature value collected at the $j$-th time point in the internal model, and $u(k+ij)$ describes the control sequence. When $i < j$, the sum represents the prediction of the input change sequence's effect on the output before time $k$; when $I > j$, the sum represents the prediction that the output is affected by the future input sequence. For the ease of application, the fault prediction model can also be rewritten in incremental form, which is expressed as follows:

$$y_m(k + i) = y_m(k + i - 1) + \sum_{j=1}^{N} \hat{h}_j \cdot \Delta u(k + i - j) \tag{4}$$

$$\Delta u(k + i - j) = u(k + i - j - 1) \tag{5}$$

In the case where one temperature controller controls multiple sintering furnaces, the output change of the fault prediction model is calculated based on the internal model of the temperature controller controlling the sintering furnaces. The temperature change based on the fault prediction model's output is between (N-1) the temperature change of the internal model corresponding to the (N-1) sintering furnace and the temperature change of the internal model corresponding to the N sintering furnace. Then, the number of fault implementation channels is M-(N-1). M represents the number of sintering furnaces controlled by the temperature controller. A sintering furnace and the data transmission channel between the sintering furnace and the temperature controller constitute an implementation channel.

## Simulation experiment

The DCS hardware circuit is designed; an embedded operating system with good performance is transplanted according to functional requirements, improving development efficiency. An embedded sintering furnace temperature control system is designed, including the following six tasks: temperature acquisition, output control, control strategy, button setting, terminal display, and fault alarm. The experimental device includes the console (LPC1788 control platform and PC) and the controlled object (heating furnace, temperature sensor, and power regulator).

## Results and discussion

### Result of sintering furnace temperature control based on DCS control system

The furnace temperature is acquired by the temperature sensor of the thermistor PT100, and the collected real-time temperature is converted into a standard current signal through the A/D conversion module. Finally, the corresponding temperature is displayed. The designed embedded sintering furnace's temperature control system can control the temperature in two modes: manual and automated. The mode switch button in the parameter adjustment interface can adjust the temperature increase or decrease, completing the precise control of the temperature. Finally, the requirements on the temperature control function are met through continuous debugging. The accuracy is controlled to ±1˚C. The objects of this experiment are 1# and 2# sintering furnaces. After the parameters are set, the program is run, and the online monitoring is started. The curves of the sintering furnace during the heating process are shown in Fig 8. The real-time temperature of the upper, middle, and lower currents of 1#

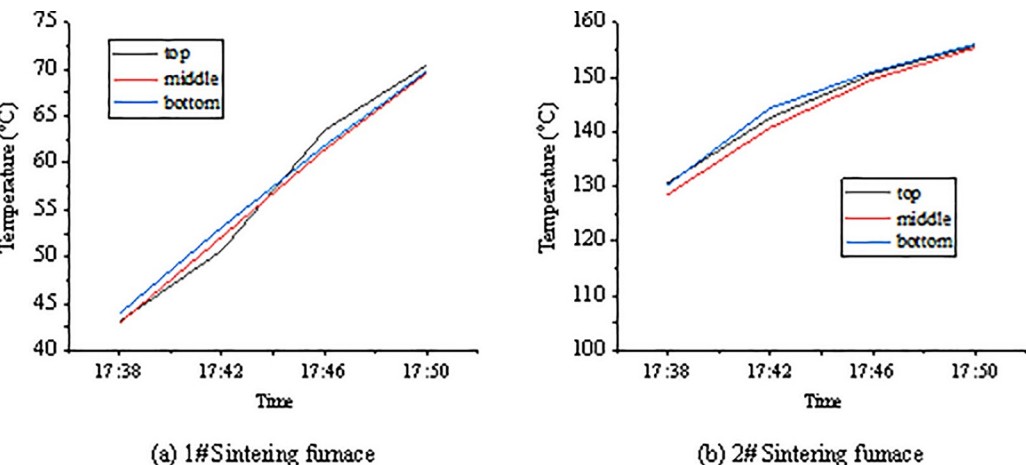

(a) 1#Sintering furnace          (b) 2# Sintering furnace

**Fig 8. Real-time temperature curves during the heating of sintering furnace.**

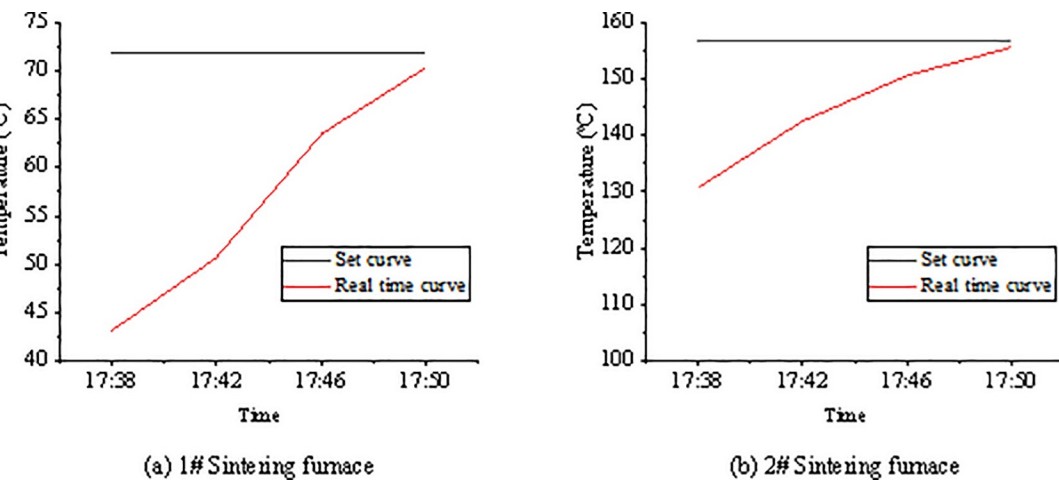

**Fig 9. Temperature control curves obtained under the DCS system.**

sintering furnace at a particular point are 56.95˚C, 56.58˚C, and 57.2˚C, respectively. The real-time temperature of the upper, middle, and lower currents of 2# sintering furnaces at a particular point is 144.7˚C, 143.8˚C, and 144.0˚C, respectively. Three currents' overall temperature control deviation stays in the controllable range in the two sintering furnaces, which meets the design requirements. The temperature control curve interface displays the real-time temperature and the trend of temperature changes. The temperature control curve obtained under DCS is presented in Fig 9.

## Results of sintering furnace fault diagnosis

Faults of the sintering furnaces are simulated, including the sensor faults, the communication channel faults, and the implementation channel faults. The fault diagnosis needs to be based on data analysis. If the monitored sintering furnace data come from the scene, the data are collected and stored in the server database through the receiving module; then, the data are distributed to the online monitoring and fault diagnosis module by Ethernet. The final diagnosis result is obtained and stored in the information database. If the monitored sintering furnace data come from the database model, the data will be taken out of the offline modeling database directly at the set time to simulate the on-site collection for online monitoring. The 2# sintering furnace is taken as an example, and the online monitoring results obtained are shown in Fig 10.

According to the obtained online monitoring results of the sintering furnace, the fault cause of the sintering furnace can be determined by comparing the monitoring value with the upper and lower limits. Five sets of data are used to diagnose the abnormal temperature of the sintering furnace through the fuzzy expert system. The proposed diagnosis method is compared with the method based on the BP neural network to prove its reliability. The results are presented in Fig 11. BP neural network takes the fault factor as input to construct the neural network prediction model. Because the data structure of the fault factor of the sintering furnace is complicated, the final fault diagnosis accuracy is lower than that of the fuzzy expert system. During the fault diagnosis process, each sub-fault diagnosis is equivalent to a sub-expert system. The entire fault diagnosis expert system and its sub-systems are integrated to construct a complicated process control integration model to achieve the distributed fault diagnosis.

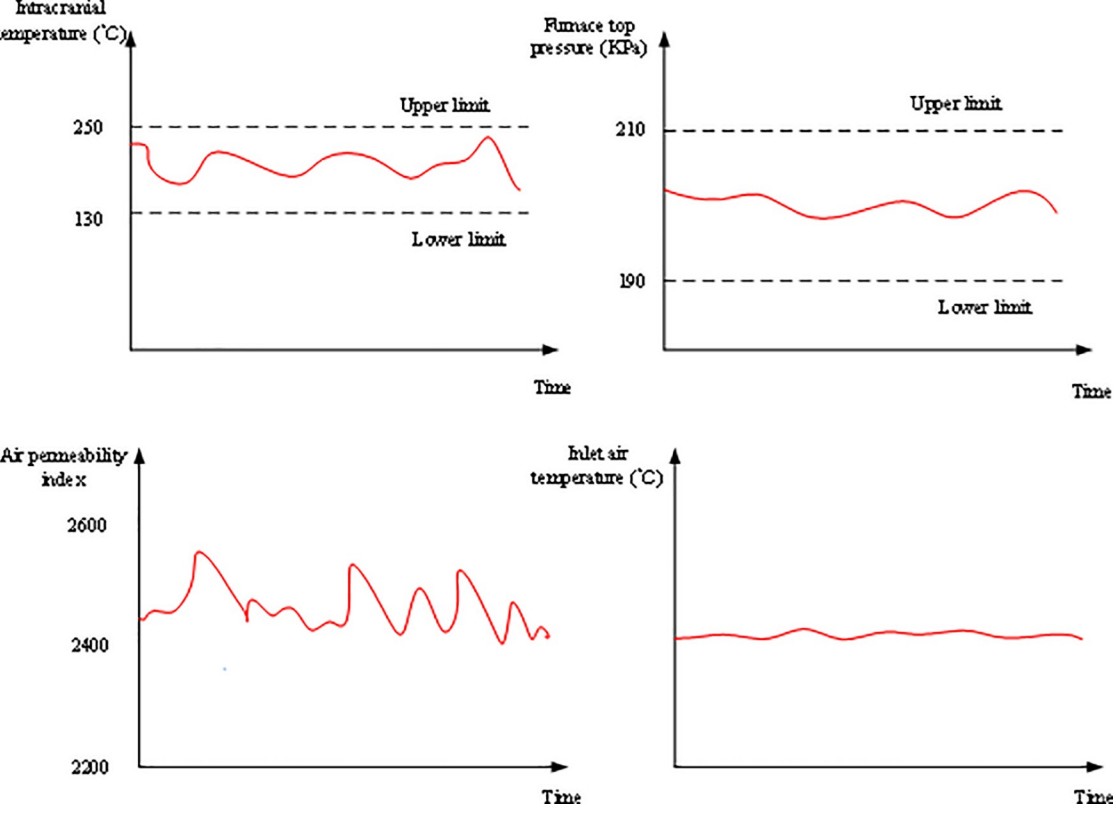

**Fig 10. Online monitoring results of 2# sintering furnace.**

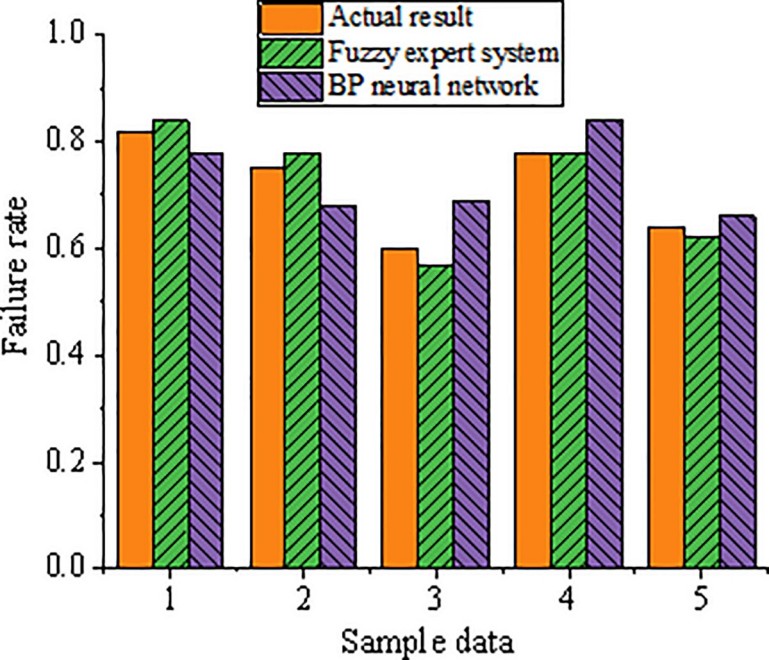

**Fig 11. Comparison of two fault detection methods for sintering furnace.**

## Discussion

Modern industry requires real-time control of production and processing equipment and components for fault diagnosis and energy saving. Edge computing technology is introduced to solve cloud servers' limited processing capacity in the cloud computing-based intelligent equipment architecture so that industrial equipment can respond and process quickly without connecting to the cloud. The structure and working model of the sintering furnace in the metallurgical industry are analyzed after discussing intelligent industrial equipment feasibility. Temperature control is the vital link to the sintering furnace control system. According to the sintering furnace's technological characteristics, DCS, PLC controller, and configuration software technology are organically combined. Then, the embedded thinking is introduced, and a temperature control system of sintering furnaces suitable for modern industrial production is developed. In the meantime, the sintering furnace temperature DCS is applied to achieve fault self-diagnosis and fault tolerance control in the sintering furnace system through the integrated software algorithm. At present, traditional regular maintenance and fault diagnosis solutions cannot effectively detect potential failures of industrial equipment. When machine learning technology is applied to data analysis and fault diagnosis, the original data will first be denoised. The denoised data are sparsely coded to train the dictionary, and the final potential faults will be identified and classified by the Support Vector Machine (SVM) [33]. This is also a valuable research direction for fault detection in industrial production systems.

To verify the reliability of the designed sintering furnace temperature control system, tests on two sintering furnaces reveal that the DCS-based temperature control system can meet the functional requirements and control the accuracy to within ±1˚C. The three currents' overall temperature control deviation in the two sintering furnaces is within the controllable range, meeting the design requirements. In the fault diagnosis system for sintering furnaces, fuzzy logic is introduced into the sintering furnace's parameter-changing process to comprehensively predict the status of the sintering furnaces combined with expert domain knowledge rules. The final fault diagnosis accuracy is significantly improved, which is closer to the actual situation compared with the fault diagnosis method based on BP neural network. Yavuz and Cumali [34] transferred the temperature and relative humidity data in industrial automation to Matlab/Simulink through PLC. Then, they used PLC to communicate with the microcontroller and obtain the required output in the best possible way. On this basis, PLC and DCS are combined in the present work to design the temperature control system.

## Conclusion

In the present work, the controlling idea of the edge computing intelligence agent is introduced based on the DCS control system; then, PLC is utilized to automatically control the temperature of the sintering furnace. The results obtained have significant value for the following research on intelligent control in industrial processing and production. Nevertheless, there are several shortcomings in the research. The temperature field simulation is performed in an ideal state. In reality, the temperature field in the sintering furnace should consider the external environment's interference factors, which will be discussed in the future.

## Supporting information

**S1 Data.**
(ZIP)

## Author Contributions

**Conceptualization:** Na Qu.

**Data curation:** Na Qu.

**Formal analysis:** Na Qu.

**Funding acquisition:** Na Qu.

**Investigation:** Na Qu.

**Methodology:** Na Qu.

**Project administration:** Na Qu.

**Resources:** Wen You.

**Software:** Wen You.

**Supervision:** Wen You.

**Validation:** Wen You.

**Visualization:** Wen You.

**Writing – original draft:** Wen You.

**Writing – review & editing:** Wen You.

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
