## [Decision Letter · Decision Letter 0]

18 Apr 2021

PONE-D-21-10646

Design and Fault Diagnosis of DCS Sintering Furnace’s Temperature Control System for Edge Computing

PLOS ONE

Dear Dr. You,

Thank you for submitting your manuscript to PLOS ONE. After careful consideration, we feel that it has merit but does not fully meet PLOS ONE’s publication criteria as it currently stands. Therefore, we invite you to submit a revised version of the manuscript that addresses the points raised during the review process.

Based on the comments received form the reviewers and my own observation, I recommend major revisions for the paper.

We look forward to receiving your revised manuscript.

Kind regards,

Thippa Reddy Gadekallu

Academic Editor

PLOS ONE

Journal Requirements:

PLOS requires an ORCID iD for the corresponding author in Editorial Manager on papers submitted after December 6th, 2016. Please ensure that you have an ORCID iD and that it is validated in Editorial Manager. To do this, go to ‘Update my Information’ (in the upper left-hand corner of the main menu), and click on the Fetch/Validate link next to the ORCID field. This will take you to the ORCID site and allow you to create a new iD or authenticate a pre-existing iD in Editorial Manager. Please see the following video for instructions on linking an ORCID iD to your Editorial Manager account: https://www.youtube.com/watch?v=_xcclfuvtxQ

Please include captions for your Supporting Information files at the end of your manuscript, and update any in-text citations to match accordingly. Please see our Supporting Information guidelines for more information: http://journals.plos.org/plosone/s/supporting-information.

Reviewers' comments:

Reviewer's Responses to Questions

**Comments to the Author**

1. Is the manuscript technically sound, and do the data support the conclusions?

Reviewer #1: Yes

Reviewer #2: Yes

Reviewer #3: Yes

2. Has the statistical analysis been performed appropriately and rigorously? 

Reviewer #1: Yes

Reviewer #2: Yes

Reviewer #3: Yes

3. Have the authors made all data underlying the findings in their manuscript fully available?

Reviewer #1: Yes

Reviewer #2: Yes

Reviewer #3: Yes

4. Is the manuscript presented in an intelligible fashion and written in standard English?

Reviewer #1: No

Reviewer #2: No

Reviewer #3: Yes

5. Review Comments to the Author

Reviewer #1: 1. Gap in the article is not presented properly by the authors.

2. Author failed to highlight contributions in this article

3. The limitations of the existing works which prompted the authors to carry out the current research can be discussed in introduction.

4. Related works section can be summarized as a table.

5. Figures are not clear. Unable to see the text in figures. Redraw the figures.

6. How is the paper different from existing works? What gaps they identified in existing works? How the proposed approach solves the research/scientific problems identified in the existing works?

7. Authors should tabulate the values obtained for the performance measures used to evaluate their contribution.

Reviewer #2: 1. The authors have to carefully proofread the paper to improve it grammatically.

2. WHat are the limitations of the existing works?

3. List out the main contributions of the current work.

4. Compare the current work with recent state-of-the-art.

5. Discuss how machine learning can be useful for fault diagnosis. The authors can refer these articles"An adaptive multi-layer botnet detection technique using machine learning classifiers, A novel ensemble of hybrid intrusion detection system for detecting internet of things attacks" which use ML for different applications.

6. Present a detailed analysis on the results obtained.

Reviewer #3: 1. Abstract add the results achieved

2. In introduction add the contributions

3. Add the latest references

4. figures quality has to be improved

5. The authors can cite the following references

Fusion of Federated Learning and Industrial Internet of Things: A Survey

A survey on blockchain for big data: Approaches, opportunities, and future directions

6. PLOS authors have the option to publish the peer review history of their article (what does this mean?). If published, this will include your full peer review and any attached files.

Reviewer #1: **Yes: **KADIYALA RAMANA

Reviewer #2: No

Reviewer #3: No

---

## [Author Response · Author response to Decision Letter 0]

26 May 2021

Reviewer #1: 

1. Gap in the article is not presented properly by the authors.

Response: Thanks for reviewing this article. A “Related Work” section has been supplemented to present the research gaps.

2. Author failed to highlight contributions in this article

Response: Contributions have been highlighted in the introduction section. 

3. The limitations of the existing works which prompted the authors to carry out the current research can be discussed in introduction.

Response: A “Related Work” section has been supplemented to discuss the limitations of the existing works which prompted the current research.

4. Related works section can be summarized as a table.

Response: A “Related Work” section has been supplemented.

5. Figures are not clear. Unable to see the text in figures. Redraw the figures.

Response: The figures have been redrawn.

6. How is the paper different from existing works? What gaps they identified in existing works? How the proposed approach solves the research/scientific problems identified in the existing works?

Response: In the supplemented “Related Work” section, the differences between the paper and existing works have been explained; besides, the contributions of this paper have been highlighted in the conclusion section.

7. Authors should tabulate the values obtained for the performance measures used to evaluate their contribution.

Response: The system test results are indicators for evaluating the contribution of the paper, which have been shown through figures.

Reviewer #2: 

1. The authors have to carefully proofread the paper to improve it grammatically.

Response: Thanks for your comment. The paper has been proofread.

2. WHat are the limitations of the existing works?

Response: A “Related Work” section has been supplemented to discuss the limitations of the existing works which prompted the current research.

3. List out the main contributions of the current work.

Response: The contributions of this paper have been highlighted in the conclusion section.

4. Compare the current work with recent state-of-the-art.

Response: The current work has been compared with recent state-of-the-art. 

5. Discuss how machine learning can be useful for fault diagnosis. The authors can refer these articles"An adaptive multi-layer botnet detection technique using machine learning classifiers, A novel ensemble of hybrid intrusion detection system for detecting internet of things attacks" which use ML for different applications.

Response: This article has been cited (reference [33]).

6. Present a detailed analysis on the results obtained.

Response: A detailed analysis on the results obtained has been presented.

Reviewer #3: 

1. Abstract add the results achieved

Response: Thanks for reviewing this paper. The results achieved have been added in the abstract. 

2. In introduction add the contributions

Response: Contributions have been added in the introduction section.

3. Add the latest references

Response: Latest references have been cited in this revision.

4. figures quality has to be improved

Response: Figures have been redrawn. 

5. The authors can cite the following references

Fusion of Federated Learning and Industrial Internet of Things: A Survey

A survey on blockchain for big data: Approaches, opportunities, and future directions

Response: As you have suggested, these works have been cited (reference [3] and [5]).

---

## [Decision Letter · Decision Letter 1]

1 Jun 2021

Design and Fault Diagnosis of DCS Sintering Furnace’s Temperature Control System for Edge Computing

PONE-D-21-10646R1

Dear Dr. You,

We’re pleased to inform you that your manuscript has been judged scientifically suitable for publication and will be formally accepted for publication once it meets all outstanding technical requirements.

Kind regards,

Thippa Reddy Gadekallu

Academic Editor

PLOS ONE

Additional Editor Comments (optional):

Reviewers' comments:

Reviewer's Responses to Questions

**Comments to the Author**

1. If the authors have adequately addressed your comments raised in a previous round of review and you feel that this manuscript is now acceptable for publication, you may indicate that here to bypass the “Comments to the Author” section, enter your conflict of interest statement in the “Confidential to Editor” section, and submit your "Accept" recommendation.

Reviewer #2: All comments have been addressed

Reviewer #3: All comments have been addressed

2. Is the manuscript technically sound, and do the data support the conclusions?

Reviewer #2: Yes

Reviewer #3: Yes

3. Has the statistical analysis been performed appropriately and rigorously? 

Reviewer #2: Yes

Reviewer #3: Yes

4. Have the authors made all data underlying the findings in their manuscript fully available?

Reviewer #2: Yes

Reviewer #3: Yes

5. Is the manuscript presented in an intelligible fashion and written in standard English?

Reviewer #2: Yes

Reviewer #3: Yes

6. Review Comments to the Author

Reviewer #2: The authors have addressed all the suggestions and comments I recommend accepting the paper in its present form.

Reviewer #3: I have gone the revised paper, the authors have addressed all my comments, paper can be accepted in the current form

7. PLOS authors have the option to publish the peer review history of their article (what does this mean?). If published, this will include your full peer review and any attached files.

Reviewer #2: No

Reviewer #3: No

---

## [Editor Report · Acceptance letter]

14 Jun 2021

PONE-D-21-10646R1 

Design and Fault Diagnosis of DCS Sintering Furnace’s Temperature Control System for Edge Computing 

Dear Dr. You:

I'm pleased to inform you that your manuscript has been deemed suitable for publication in PLOS ONE. Congratulations! Your manuscript is now with our production department. 

Kind regards, 

on behalf of

Dr. Thippa Reddy Gadekallu 

Academic Editor

PLOS ONE